# Potential of Mesenchymal Stem Cells and Their Secretomes in Decreasing Inflammation Markers in Polycystic Ovary Syndrome Treatment: A Systematic Review

**DOI:** 10.3390/medicines10010003

**Published:** 2022-12-23

**Authors:** Gunawan Dwi Prayitno, Keri Lestari, Cynthia Retna Sartika, Tono Djuwantono, Andi Widjaya, R. Muharam, Yudi Mulyana Hidayat, Dewi Wulandari, Rima Haifa, Nabilla Farah Naura, Kristin Talia Marbun, Annisah Zahrah

**Affiliations:** 1Departement of Obstetrics and Gynecology of Rumah Sakit Pusat Angkatan Darat Gatot Soebroto Jakarta, Jakarta 10410, Indonesia; 2Faculty of Pharmacy of Universitas Padjajaran, Bandung 45363, Indonesia; 3PT. Prodia STEMCell Indonesia, Jakarta 10430, Indonesia; 4Faculty of Obstetrics and Gynecology of Universitas Padjajaran, Bandung 45363, Indonesia; 5Faculty of Obstetrics and Gynecology of Universitas Indonesia, Jakarta 10430, Indonesia

**Keywords:** polycystic ovary syndrome, inflammation, insulin resistance, stem cells, conditioned medium

## Abstract

**Background:** Polycystic ovary syndrome (PCOS) is a chronic disorder and is one of the most common endocrine disorders in women of a reproductive age. The prevalence of PCOS is growing globally; 52% of women in Southeast Asia alone suffer from this disorder. This disorder is caused by chronic hyperandrogenism, which hinders folliculogenesis. There is also a close relationship between hyperandrogenism and hyperinsulinemia/insulin resistance (IR), and it is estimated that 40–80% of PCOS patients suffer from insulin resistance (IR). Mesenchymal stem cells (MSCs) and their secretomes have been shown to alleviate PCOS symptoms by decreasing IR and androgen secretion by reducing inflammation. This study aimed to systematically review the literature to study the reported potential of MSCs and their secretomes in decreasing inflammation markers in PCOS treatment. **Methods:** A systematic literature search was performed on EMBASE, PubMed (MEDLINE), and the Cochrane Library with the terms *insulin-resistant PCOS*, *mesenchymal stem cells*, and *secretome* or *conditioned medium* as the search keywords. A total of 317 articles were reviewed. Four articles were identified as relevant for this systematic review. **Results:** The results of this study supported the use of mesenchymal stem cells and their secretions in decreasing inflammatory markers in the treatment of polycystic ovary syndrome. **Conclusions:** This review provided evidence that treatment with mesenchymal stem cells and their secretomes has the potential to treat PCOS due to its ability to downregulate androgen levels and increase insulin sensitivity, which thereby lowers the level of proinflammatory factors.

## 1. Introduction

Polycystic ovary syndrome (PCOS) is a chronic disorder that causes weight gain, infertility, and depression in women of a reproductive age and is one of the most common endocrine disorders in the world. PCOS occurs when the follicles do not ovulate, and the accumulation of immature follicles in the ovary forms a polycystic ovary. Fifty-two percent of women suffer from this disorder in Southeast Asia alone [1]. Women are diagnosed with PCOS if they fulfill 2 out of 3 of the following criteria: oligomenorrhea/amenorrhea, hyperandrogenism, and the presence of polycystic ovaries in an ultrasound [2]. There is a close relationship between hyperandrogenic conditions and hyperinsulinemia/insulin resistance. The prevalence of insulin resistance in PCOS is 40–80%. This insulin resistance can develop further into type 2 diabetes mellitus (DM type 2) metabolic syndrome, obesity, hypertension, coronary heart disease, stroke, kidney failure, and chronic anovulatory cycles. The initial cause of PCOS is due to micro-RNA (miRNA) dysregulation, and miRNA dysregulation is induced by the accumulation of adipose tissue, which is why obesity and weight gain are often related to PCOS. miRNAs act as post-transcriptional regulators and can act upon hundreds of target miRNAs. An imbalance in miRNA levels could cause the change and affect ovarian insulin sensitivity, hormone synthesis, and inflammation, which leads to the infant stage of PCOS [3].

PCOS is caused by an increase in the gonadotropin-releasing hormone (GnRH) signal from the hypothalamus due to a lack of adequate progesterone levels during the luteal phase [4]. Note that progesterone has a negative feedback loop on the gonadotropin-releasing hormone (GnRH) to reduce its level and keep its frequency in the normal range, and a normal GnRH level keeps the luteinizing hormone (LH) and follicle-stimulating hormone (FSH) at the correct ratio. If a follicle does not ovulate, a corpus luteum will not be created, and there will no surge in progesterone. Progesterone is a powerful GnRH inhibitor, and a lack of it leads to a higher pulse frequency of GnRH. A high GnRH level leads to an increase in LH over FSH since LH is more responsive to a faster GnRH pulse frequency. A low FSH level is bad for follicular maturation since the follicles would not be able to mature fully. An increase in LH stimulates the thecal cells to produce androgen, and this causes hyperandrogenism, which is associated with hirsutism and alopecia. High levels of androgen also reduce insulin sensitivity, which is why insulin resistance (IR) is usually linked to PCOS [4].

PCOS treatment varies depending on the severity and the comorbid condition of the patients. The standard treatment for PCOS addresses the menstrual irregularity and hyperandrogenism with oral contraceptive pills. At the same time, the metabolic features are managed with metformin, a drug used to reduce the blood sugar levels in T2DM patients [5]. Though these drugs are effective, the medication must still be taken regularly and may present mild to severe side effects. These drugs do not treat the problem at its root and only alleviate symptoms. Thus, a treatment plan that treats PCOS and has a long-term impact is needed. 

Mesenchymal stem cell (MSC)-based therapy has shown promise as a treatment option for PCOS due to its self-renewal properties, differentiation potentials, and immunomodulatory activities, especially in inflammatory-related diseases. Several studies have shown that MSCs have the potential to help restore and enhance ovarian function, which is mediated by the paracrine signaling pathways. The paracrine activity of MSCs has a more significant effect on function, as the MSC activity is regulated by the RAP1/NFkb signaling pathway, which can regulate immune and inflammatory responses, repair injured tissues, and develop progenitor cells to differentiate into tissue cells. MSCs consist of various sources, one of which is the umbilical cord. UC-MSCs are a rich source of MSCs that express stem-cell-specific markers and can be differentiated into several types of mesodermal cells for tissue repair and the regulation of immune responses [6]. The studies show that there is chronic inflammation in PCO women and, thus, that PCO women have risk factors for insulin resistance, type 2 diabetes (DBT2), metabolic syndrome, and obesity [7]. Research suggests that UC-MSC transplantation has the potential to ameliorate the pathological changes in PCOS, to restore ovarian function, and to maybe help control dehydroepiandrosterone (DHEA)-induced ovarian-related disorders. This effect is mediated by the downregulation of the expression of inflammatory cytokines, namely interleukin 1 beta (IL-1β), tumor necrosis factor alpha (TNF-α), and interferon gamma (IFN-γ), as well as fibrosis-associated genes, such as the connective tissue growth factor (CTGF) [6].

In addition, the use of secretomes may promote the effectiveness and the impact of this therapy. Secretomes may mediate the paracrine mechanism within the stem cells, thereby prolonging the impact timespan of MSCs. The secretome is an extract composed of soluble proteins, lipids, nucleic acids, and extracellular vesicles (EVs are from the medium where the stem cells are cultured). Studies have shown that secretomes have demonstrated immunosuppressive effects on chronic inflammatory diseases, such as rheumatoid arthritis, systemic lupus erythematosus, and inflammatory bowel disease [8]. Studies have shown that the treatment effects of MSCs are related to their secretomes, which support the restoration of ovarian function because these molecules include the insulin-like growth factor (IGF), vascular endothelial growth factor (VEGF), and other growth factors that induce cell growth, differentiation, and immunoregulation [6]. In addition to secretomes, there are exosomes, which are MSC-derived exosomes that deliver proteins, lipids, and miRNAs to recipient cells and regulate the inflammation of injured tissues. Exosomes can mediate anti-inflammatory activity and have a communication role between MSCs and the ovarian microenvironment [9].

Currently, the available studies provide an opportunity to summarize the potential of MSCs and their secretomes in experimental models of insulin-resistant PCOS and to gain a better understanding of how treatment-related factors influence insulin-resistant PCOS outcomes. Thus, this systematic review aimed to evaluate the therapeutic effect of MSCs and their secretomes in PCOS therapy. We hypothesized that MSCs or their secretome application in treatment would decrease inflammatory activities in PCOS patients.

## 2. Material and Methods

### 2.1. Eligibility Criteria

The inclusion criteria for this systematic review were as follows: Study design: human and controlled animal models.Study group: women and animal models of polycystic ovary syndrome.Interventions: any application of mesenchymal stem cell and stem cell-derived EVs/MVs or exomes to the study group.Outcomes: decrease of inflammation marker and regulation of FSH, LH, E2, and testosterone.Language: English.

Articles that were categorized as duplicates, review articles, non-English articles, or irrelevant articles were excluded. 

### 2.2. Literature Search and Study Selection

The Preferred Reporting Items for Systematic Reviews and Meta-Analyses (PRISMA) were used as a guide for sorting available literature for this review [10]. A systematic search was performed on PubMed, EMBASE, the Cochrane library, and ScienceDirect with keywords such as “insulin-resistant PCOS”, “PCOS-insulin resistant MSCs therapy”, “PCOS-insulin resistant stem cell therapy”, and “PCOS-insulin resistant secretome therapy”. A similar search was performed on EMBASE, the Cochrane library, and ScienceDirect. All the studies were first screened based on their title and abstract, and then full-text versions of the articles were screened for further evaluation. We limited the publication date to 2017 and April 2022. The strategy used for all databases can be seen in Figure 1.

### 2.3. Methodological Quality Assessment and Risk of Bias

A quality assessment was conducted on four included studies. The quality of three animal-based studies was assessed using Animal Research: Reporting of In Vivo Experiments (ARRIVE) guidelines. One human-based study was assessed using Cochrane risk-of-bias tool for randomized trials (RoB 2). Four authors (G.D.P., R.H., N.F.N., and K.T.M.) performed all the assessments independently. Any disagreements were resolved through discussion with other authors.

### 2.4. Data Extraction and Synthesis 

Four authors (G.D.P., R.H., N.F.N., and K.T.M.) independently analyzed and tracked the data from every included article. Disagreements between authors were resolved through discussion and scientific reasoning. The following data were extracted based on study design, type of MSC source, cell preparation method, interventions, comparison, duration of follow-up, main outcome for the studies that used MSCs as their therapy, main outcome for the studies that used secretomes as their therapy, significant results compared to control or baseline, and other outcomes. 

The data collection of in vivo study outcomes is shown in Table 1. 

## 3. Results

### 3.1. Study Selection 

A PRISMA flow diagram summarized the process of study selection [10]. A total of 317 studies were identified from the literature. After the screening of the titles and abstracts, 21 articles were eligible for further evaluation. After the full-text assessment, 4 studies were included in this systematic review, and 17 studies were excluded because they were marked as duplicates and, therefore, did not meet the inclusion criteria (KT1). After the full-text assessment, only four studies were included in this systematic review.

### 3.2. Assessment of Methodology Quality

#### Assessment of Risk of Bias

Study characteristics:

Table 1 shows the overview of the studies used in this review, and the detailed explanation of the studies can be seen in Table 2, which explains the types of MSCs used in each study, the test subjects, and the interventions used. 

The Kalhori et al., Chugh et al., and Zhao et al. studies used mice as their test subjects, while Zhao et al. used human subjects. The mice that were used as test subjects were all PCOS-induced with drugs. One study discussed the sole use of MSCs in treating PCOS, and four studies discussed the use of secretomes/exosomes/cm in treating PCOS. 

The BM-MSC type was the one that was most commonly used (two studies) followed by the umbilical cord type (one study) and the adipose tissue type (one study). Four studies used human sources of MSCs, and one study uses BM-MSCs from mice. The Kalhori et al. study used mice-derived BM-MSCs in its trial. 

The Zhao et al. and Zhao et al. studies isolated exosomes from the CM, and each journal had its respective method for isolating the exosomes [9,11]. Exosomes were all isolated from Dulbecco’s Modified Eagle’s Medium (DMEM) along with other additional reagents, such as 10% fetal bovine serum (FBS) and other antibiotics [9,11,12,13]. The delivery method of the exosomes in the two studies was via injection as in all the studies [9,11,12,13].

### 3.3. In Vivo Study Outcomes

The in vivo study outcomes of the four articles used are summarized in Table 2, which includes the main outcome measures, the scores and results of the main groups, the statements of statistical significance, other parameters, and a list of other outcome measures.

All four studies targeted the ovary overall; though, each had its specific target. The Kalhori et al. study’s scope was the widest compared to the others since it measured the oxidative stress hormone, an inflammatory cytokine, and reproductive hormones [11]. Zhao et al. focused on targeting the cumulus cells instead of the overall ovary. The other study by Zhao et al. only focused on the activity of miR-323-3p in the PCOS patients, a type of micro-RNA (mi-RNA) that contributes to the regulation of inflammatory and immune responses. All the studies were supported by pre-in vitro studies, which oversaw the increased proliferation, viability, and migration levels in the CM group, strengthening the prior claim.

The functional outcomes were reported in all five of the studies, albeit with different parameters and procedures. Kalhori et al. reported a significant increase in FSH and tacrolimus (TAC) and a significant decrease in testosterone, LH, TUNEL-positive apoptotic cells, and AMH. The Chugh et al. results focused on the impact of IL-10, which was found to be the factor that was responsible for providing an anti-inflammatory effect and regulating steroidogenic gene expression in the PCOS mice. Zhao et al. reported the anti-inflammatory effect presented by hUC-MSCs specifically on the granulosa cells in the follicular fluid of the PCOS patients, and it was discovered that tumor necrosis factor alpha (TNF-α) and interferon gamma (IFN-γ) were significantly decreased and that IL-10 levels were upregulated in the PCOS patients [9]. Another study by Zhao et al. focused on measuring the adipose mesenchymal stem cells’ (AMSC) efficacy on cumulus cells (CCs) and discovered that miR-323-3p, a micro-RNA (miRNA) responsible for cell proliferation and apoptosis suppression, was upregulated in the PCOS mouse models [12].

## 4. Discussion

### 4.1. Principal Findings

Based on the selected journals, it could be concluded that MSCs and their secretomes can promote anti-inflammation and antiapoptosis activity in IR-PCOS subjects. MSCs and their secretomes suppressed inflammation by targeting oxidative stress pathways and inflammation pathways and by increasing insulin sensitivity [11]. 

The use of MSCs in treating inflammatory diseases has been well established. However, the use of secretomes in immunotherapy is not as well known, which is shown by the limited amount of literature on the uses of the secretome [14]. Secretomes secreted cytokines and other factors to aid in the anti-inflammation response. However, they presented a lower efficacy than MSCs since they had no proliferative capability (limited with a half-time). Zhao et al. and Zhao et al. measured the effectiveness of exosomes in treating IR-PCOS patients. Both studies successfully suppressed the granulosa cells’ apoptosis through their respective mechanisms [9,11]. Due to the heterogeneity of the parameters, it was hard to compare the results of the MSC-focused and secretome-focused studies. 

MSCs and their secretomes presented higher functional fertilities than the control groups based on the change in the levels of inflammation, oxidative stress, and fertility markers [9,11,12,13]. The markers of inflammation included high sensitivity C-reactive protein (hs-CRP), IL-6, IL-10, plasminogen activator inhibitor I (PAI-I), TNF-α, IFN-γ; the markers of oxidative stress included malondialdehyde (MDA) and the total antioxidant capacity (TAC); and the markers of fertility included the anti-Mullerian hormone (AMH), the follicle-stimulating hormone (FSH), the luteinizing hormone (LH), estrogen, progesterone, and testosterone. 

IR played an important role in the inflammatory process. The probability that IR causes PCOS was determined through the markers of inflammation in both IR and PCOS. It was discovered that IR had the same inflammation and oxidative stress markers seen in PCOS: hs-CRP, IL-6, PAI-I, TNF-α, and IFN-γ [15]. Among the four selected studies, only Kalhori et al. measured the correlation between insulin sensitivity and the oocyte volume. It was observed that IR, androgen secretion, and the fat metabolism were all linked to inflammation. 

#### 4.1.1. Target Pathways in PCOS Treatment

Inflammation, hyperandrogenism, and IR are the three leading causes of PCOS; all three reasons are interrelated, and each cause affects the others directly. Since all three causes are related, targeting one cause will affect the others. 

In terms of inflammatory activity, there was a significant decrease in proinflammatory cytokines (IL-6 and TNF-α) and MDA and an increase in the total antioxidant capacity upon MSC/exosome injection [11,13]. Kalhori et al., Chugh et al., and Zhao et al. stated that, aside from a decrease in proinflammatory cytokines, there was an increase in IL-10, an inflammatory cytokine [9,12,13]. This finding was aligned with previous studies that reported MSCs’ anti-inflammatory and immunomodulatory effects on the innate and adaptive immune system, an inhibition of proinflammatory cytokine expression (IFN-γ, TNF-α, IL-6, and IL-17), and a promotion of anti-inflammatory expression (IL-4 and IL-10) [16]. A decrease in the inflammatory response will cause a reduction in the PCOS symptoms presented.

Inflammation was linked with hyperandrogenism since proinflammatory cytokines (e.g., TNF-α) stimulated the proliferation of androgen-producing theca cells. The effect of MSCs on steroidogenesis could be justified by the positive impact mentioned by Kalhori et al., Chugh et al., and Zhao et al. [9,12,13] Each study discussed the significant reduction in testosterone after MSC injection. There was an improvement in the disorders and dysfunction related to hyperandrogenism, such as a reduction in the primary, preantral, and atretic follicles [11]. The decrease in follicles meant that the egg had successfully matured and that the granulosa cells could develop due to increased progesterone. There might have been an incidence whsere the steroid hormone levels (e.g., testosterone) did not change despite the MSC treatment. This may be due to the episodic nature of steroid hormone secretion [12]. Thus, although the testosterone level measurement was the optimal measurement of androgen reduction posttreatment, we could also measure a change in steroidogenesis activity by measuring the increase in granulosa cells. 

The route of administration for MSCs and secretomes/exosomes in one report was via a subcutaneous injection (tail vein), in another report was via an intraovarian injection, and in two reports was via the granulosa cell medium of the follicular fluid of the PCOS patients undergoing IVF. From the three methods of administering MSCs and secretomes/exosomes, it could be seen that the mechanisms of action in the endocrine and paracrine pathways were all significant in reducing proinflammatory cytokines (IL-6, TNF alpha, and IFN gamma) and increasing anti-inflammatory cytokines (IL-10), the FSH hormone (serum/intraovarian), and the folliculogenesis/pregnancy rate. Changes in the regulation of the immune system and inflammation were controlled by noncoding RNA (miRNA). From the four reports, only one report examined miRNA, and the results showed that there was an upregulation of miRNA 323-3p [9,11,12,13].

#### 4.1.2. Efficacy of MSC Treatment in PCOS Patients

The fertility-related hormones were measured post-MSC/secretome injection, and the four studies showed an increase in estrogen and progesterone and a decrease in the androgen-related hormones. Kalhori et al., Chugh et al., and Zhao et al. presented an increase in fertility, with a significant increase in the ovary volume, cortex, number of antral follicles, volume of oocytes, and zona pellucida thickness and a significant decrease in the primary and preantral follicles numbers in the PCOS + BM-MSC group in comparison with the PCOS group. 

All four selected studies stated that there was a significant decrease in the inflammation markers after the MSC treatment for PCOS. Additionally, some studies also discussed the correlation between insulin resistance and PCOS, and some reported that insulin sensitivity increased after the MSC treatment. Thus, we predicted that MSCs and their secretomes could treat IR, which in turn would cause an improvement in PCOS symptoms since the factors that act as markers of inflammation would no longer be produced. 

IL-10 is an anti-inflammatory molecule that was reported in all five studies as being responsible for suppressing the inflammatory marker molecules (IL-6, TNF-α, PAI-I, and hs-CRP). All four articles conducted an ovarian stereological test including FSH, LH, TNF-α, testosterone, and IL-6. There was a decrease in the inflammatory marker levels and an increase in the fertility-related hormones following the administration of MSCs and their secretomes. 

However, this review came with several limitations. The first limitation was the different MSC sources and levels of conditioned media (secretomes or exosomes). The second was the type of PCOS inducer, which caused PCOS severity level heterogeneity in the mice and human test subjects. The third and final one was the various assessment parameters, making it unfeasible for a quantitative analysis to be conducted. Since a meta-analysis could not be conducted, more articles and research using a standard parameter should be carried out. 

## 5. Conclusions

The use of MSCs and their secretomes has potential in treating PCOS patients as shown by the available clinical and preclinical studies. Furthermore, this treatment could also be applied to diabetic patients, as IR is the first mechanism to be targeted. However, there needs to be more preclinical and clinical trials to confirm the benefits of MSC and secretome therapy and to know the most suitable MSC source and CM for future clinical studies.

## Figures and Tables

**Figure 1 medicines-10-00003-f001:**
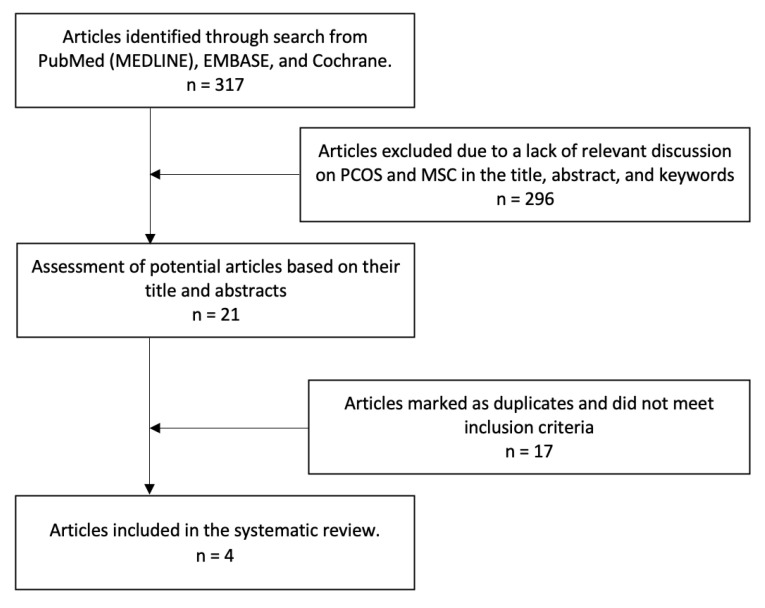
Flowchart of study selection process.

**Table 1 medicines-10-00003-t001:** Overview of the studies.

Author	Type of MSC (Donor)	Stem Cell Parts	Test Subjects	Test Subject/Cells Preparation	Intervention in the Main Group	Control(s)
Kalhori et al. [11]	Bone Marrow Mesenchymal Stromal Cells (BM-MSCs) (mouse)	-	Mouse	PCOS in mice was induced through daily subcutaneous injections of testosterone enanthate. BM-MSCs were obtained from the femurs and tibiae of 6- to 8-week-old female mice.	BM-MSCs were injected into the mice through the tail vein (10^6^ MSCs/animal) at 1 and 14 days after induction of PCOS.	Control group mice were injected subcutaneously with 0.1 mL of sesame oil.
Chugh et al. [12]	Human Bone Marrow Mesenchymal Stem Cells (IBM-MSCs) (human)	Secretome	Mouse	PCOS in mice was induced through letrozole daily for five weeks. BM-hMSCs were obtained from a 32-year-old healthy nondiabetic female donor.	BM-hMSCs: 5.0 × 10^5^ hMSCs were injected in both ovaries. Secretome group: 10 μL of concentrated secretomes were injected in both ovaries.	10 μL of PBS injected into both ovaries as a control.
Zhao et al. [9]	Human Umbilical Cord Mesenchymal Stem Cells (hUC-MSCs)	Exosome	Human	hUC-MSCs were isolated from the human umbilical cord of a healthy full-term fetus.	Women with PCOS were treated with hUC-MSC exosomes.	Healthy women were treated with hUC-MSC exosomes.
Zhao et al. [13]	Adipose Mesenchymal Stem Cells (AMSCs)	Exosome	Mouse	PCOS in mice was induced through injection of letrozole daily for five weeks. AMSCs were obtained from subcutaneous adipose tissues of four female patients (24, 34, 38, and 41 years old).	AMSCs containing miRNA-323-3p dose injections were divided into four groups: - PCOS mice that received plenty of miRNA-323-3p; - PCOS mice that received plenty of miRNA-control; - PCOS mice that received plenty of anti-miRNA-323-3p; - PCOS mice that received plenty of anti-miRNA-control.	AMSC conditioned DMEM was used as the negative control and was received through injection.

Bone marrow mesenchymal stromal cells (BM-MSCs), human bone marrow mesenchymal stem cells (IBM-MSCs), human umbilical cord mesenchymal stem cells (hUC-MSCs), human adipose mesenchymal stem cells (AMSCs), micro-RNA (miRNA), and Dulbecco’s Modified Eagle’s Medium (DMEM).

**Table 2 medicines-10-00003-t002:** In vivo study results.

Author	Assessment	Main Outcome Measure(s)	Scores/Results	Significant Difference Between Groups	Other Evaluation	Other Outcome Measure(s)
Kalhori et al. [11]	4 wk post-MSC injection of 0.1 mL	Testosterone FSH LH IL-6, TNF-α, MDA, TAC, CD31	PCOS + BM-MSC: 4.37 ± 0.47 ng/mL PCOS: 12.25 ± 0.51 ng/mL PCOS + BM-MSC: 3.24 ± 0.41 mIU/mL PCOS: 2.23 ± 0.21 mIU/mL PCOS + BM-MSC: 3.97 ± 0.35 mIU/mL PCOS: 5.84 ± 0.69 mIU/mL	In comparison between the PCOS + BM-MSC group and the PCOS control group, we found the following: - A significant increase in the serum level of FSH and TAC; - A significant decrease in the serum level of testosterone, LH, MDA, and percentage of TUNEL-positive apoptotic cells in the PCOS + BM-MSC group in comparison with the PCOS group; - A significant decrease in the serum levels of IL-6, TNF-α, and MDA and a significant increase in total antioxidant capacity in PCOS + BM-MSC group; - Localization of CD31 protein was more intense in the PCOS + BM-MSC group compared with the PCOS group.	Ovary volume, cortex, number of antral follicles, volume of oocytes, zona pellucida thickness, and primary and preantral follicles number.	A significant increase in ovary volume, cortex, number of antral follicles, the volume of oocytes, and zona pellucida thickness and a significant decrease in primary and preantral follicle numbers in PCOS + BM-MSC group were found in comparison with the PCOS group.
Chugh et al. [12]	2 wk post-BM-hMSC secretome injection of 5.0 × 105 cells	IL-10	IL-10 level was highest in BM-hMSC-treated PCOS ovaries (5.37 ± 2.72 fold) compared with untreated PCOS ovaries (1.19 ± 0.46 fold).	In comparison between the PCOS + BM-hMSC group and the PCOS control group, we found the following: - The anti-inflammatory cytokine interleukin 10 (IL-10) played a key role in mediating the effects of BM-hMSCs in PCOS model; - BM-hMSC treatment was improved in metabolic and reproductive markers in our PCOS model and was able to restore fertility; - BM-hMSCs could reverse PCOS-induced inflammation through IL-10 secretion.	Cell proliferation, steroidogenic gene expression, and fertility.	- A significant decrease in steroidogenic gene expression and curb inflammation; - Increase in fertility.
Zhao et al. [9]	48 h post-hUC-MSC-derived exosome treatment with 5 μg/mL	IL-10, TNF-α, IFN-γ	IL-10 level was highest in the hUC-MSC-derived exosome-treated group. Meanwhile, TNF-α and IFN-γ levels were the lowest in the hUC-MSC-derived exosome-treated group.	In comparison between the PCOS + hUC-MSC group and the PCOS control group, we found the following: - IL-10 was upregulated in the main group; - TNF-α and IFN-γ were downregulated. Overall, hUC-MSC-exosomes had the effects of inhibiting apoptosis of ovarian pics and promoting progesterone production.	Signaling pathway identification was responsible for anti-inflammatory activities.	UC-MSC-derived exosomes presented anti-inflammatory capabilities in pass-through inhibition of NF-κB signaling activation.
Zhao et al. [13]	5 wk post-inoculation with 5 LG/lL	miR-323-3p PDCD4	miR-323-3p level was highest in the miR-323-3p-lenti group and lowest in anti-miR-323-3p-lenti group. PDCD4 level was highest in the anti-miR-323-3p-lenti group and lowest in the miR-323-3p-lenti group.	In comparison between the PCOS + AMSC group and the PCOS control group, we found the following: - Supplementation of miR-323-3p promoted cell growth and inhibited apoptosis in vivo; - PDCD4 was upregulated in PCOS patients, and PDCD4 inhibited miR-323-3p expression; - Testosterone, FSH, and LH were downregulated in the main group, and E2 was upregulated.	Testosterone, FSH, LH, and E2.	An upregulation of serum FSH, LH, and testosterone and a downregulation of E2 levels were found in PCOS mice compared to control.

Polycystic ovary syndrome (PCOS), bone marrow mesenchymal stromal cells (BM-MSCs), human bone marrow mesenchymal stem cells (hBM-MSCs), human umbilical cord mesenchymal stem cells (hUC-MSCs), adipose mesenchymal stem cells (AMSCs), gonadotropin-releasing hormone (GnRH), luteinizing hormone (LH), follicle-stimulating hormone (FSH), estradiol (E2), interleukin 6 (IL-6), interferon γ (IFN-γ), tumor necrosis factor α (TNF-α), malondialdehyde (MDA), total antioxidant capacity (TAC), and PCOS granulosa cells (ipRGCs).

## Data Availability

We are unable to provide data to post/submit as the information is confidential to our institution.

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
