# Peer review of "Potential of Mesenchymal Stem Cells and Their Secretomes in Decreasing Inflammation Markers in Polycystic Ovary Syndrome Treatment: A Systematic Review"

_medicines, 2022, doi:10.3390/medicines10010003_

Round 1

Reviewer 1 Report

This systematic review supports that MSC and their secretome have the potential to treat PCOS . This topic is interesting with clinical significance. Firstly, the authors introduce the epidemiological characteristics, clinical manifestations and treatments of PCOS and emphasize the importance of developing treatments for PCOS. The systematic search process was logical . The four included studies provide strong evidence on potential therapeutic prospects of mesenchymal stem cells and their secretome on PCOS by down-regulating the level of androgen and pro-inflammatory factors and increasing insulin sensitivity. Moreover, the authors state the potential limitations of these kinds of studies, which provides a good idea for further research on this topic. However, the manuscript requires minor revisions.

1.The manuscript need to be carefully edited by someone with expertise in technical English editing, paying particular attention to English grammar, spelling and sentence structure so that the study is clear to the readers. For example, in page 1 ,line 14 , suffers should be suffer; in page 1 ,line 16 , dan should be and;  in page 1 ,line 17, its should be their ; in page 1, line 25, to decreased should be decrease ;  in page 1 ,line 27, increase should be increasing.

2. Please define all abbreviations in the text when used for the first time. For example, In page 2 ,line 45 miRNA; In page 2 ,line 51 GnRH.

3. In Figure 1: What dose 21articles potential articles based on their title and mean?

4. In page 8 ,line 274: “All five articles” , Please confirm if this is correct.

5. In the four included studies, MSC or their secretome were injected through the vein or in both ovaries. What are the effects of different methods of administration on the treatment of PCOS? Please state in the discussion.

Author Response

Response to Reviewer 1 Comments

This systematic review supports that MSC and their secretome have the potential to treat PCOS. This topic is interesting with clinical significance. Firstly, the authors introduce the epidemiological characteristics, clinical manifestations and treatments of PCOS and emphasize the importance of developing treatments for PCOS. The systematic search process was logical. The four included studies provide strong evidence on potential therapeutic prospects of mesenchymal stem cells and their secretome on PCOS by down-regulating the level of androgen and pro-inflammatory factors and increasing insulin sensitivity. Moreover, the authors state the potential limitations of these kinds of studies, which provides a good idea for further research on this topic. However, the manuscript requires minor revisions.

Point 1: The manuscript need to be carefully edited by someone with expertise in technical English editing, paying particular attention to English grammar, spelling and sentence structure so that the study is clear to the readers. For example, in page 1, line 14, “suffers” should be “suffer”; in page 1, line 16, “dan” should be “and”; in page 1, line 17, “its” should be “their”; in page 1, line 25, “to decreased” should be “decrease”; in page 1, line 27, “increase” should be “increasing”.

Response 1: Please provide your response for Point 1. (in red)

Point 2: Please define all abbreviations in the text when used for the first time. For example, In page 2, line 45 “miRNA”; In page 2, line 51 “GnRH”.

Response 2: Please provide your response for Point 2. (in red)

Point 3: In Figure 1: What dose “21articles potential articles based on their title and” mean?

Response 3: Please provide your response for Point 3. (in red)

Point 4: In page 8, line 274: “All five articles”, Please confirm if this is correct.

Response 4: Please provide your response for Point 4. (in red)

Point 5: In the four included studies, MSC or their secretome were injected through the vein or in both ovaries. What are the effects of different methods of administration on the treatment of PCOS? Please state in the discussion.

Response 5: Please provide your response for Point 5. (in red)

Reviewer 2 Report

The authors performed a systematic review of the literature on the role of mesenchymal cells in PCOS treatment.

Major points:

1. The flowchart (figure 1) was confusing. Texts within the following boxes were incomplete: “296 articles eliminated due to title and abstract”; “21 articles potential articles…” and “15 articles marked as…”. Please make sure the flowchart can be easily understood and the information is complete and readable. However this flowchart appears to be inconsistent with lines 154-159. In the flow chart, it says “15 articles marked as duplicates and did not meet” but in the Results section, it says 17 studies were excluded. Please explain the inconsistencies between the figure and text.

Minor points:

2. Please annotate your tables with what each abbreviation stands for in the footnote. The tables need to be able to stand alone.

3. There are numerous grammatical mistakes in the current manuscript that the authors should correct.

Author Response

Response to Reviewer 2 Comments

The authors performed a systematic review of the literature on the role of mesenchymal cells in PCOS treatment.

Major points:

Point 1: The flowchart (figure 1) was confusing. Texts within the following boxes were incomplete: “296 articles eliminated due to title and abstract”; “21 articles potential articles…” and “15 articles marked as…”. Please make sure the flowchart can be easily understood and the information is complete and readable. However this flowchart appears to be inconsistent with lines 154-159. In the flow chart, it says “15 articles marked as duplicates and did not meet” but in the Results section, it says 17 studies were excluded. Please explain the inconsistencies between the figure and text.

Response 1: Please provide your response for Point 1. (in red)

Minor points:

Point 2: Please annotate your tables with what each abbreviation stands for in the footnote. The tables need to be able to stand alone.

Response 2: Please provide your response for Point 2. (in red)

Minor points:

Point 3: There are numerous grammatical mistakes in the current manuscript that the authors should correct.

Response 3: Please provide your response for Point 3. (in red)

Round 2

Reviewer 2 Report

The authors have revised and improved their manuscript based on my comments. Please make sure to include point-by-point responses in all future manuscript revisions. 

Please make sure all grammatical errors are corrected.

Author Response

Response to Reviewer 2 Comments

The authors have revised and improved their manuscript based on my comments. Please make sure to include point-by-point responses in all future manuscript revisions. 

Please make sure all grammatical errors are corrected.

Response 1: Please provide your response for Point 1. (in red)
